# Artificial Intelligence for Evaluation of Retinal Vasculopathy in Facioscapulohumeral Dystrophy Using OCT Angiography: A Case Series

**DOI:** 10.3390/diagnostics13050982

**Published:** 2023-03-04

**Authors:** Martina Maceroni, Mauro Monforte, Rossella Cariola, Benedetto Falsini, Stanislao Rizzo, Maria Cristina Savastano, Francesco Martelli, Enzo Ricci, Sara Bortolani, Giorgio Tasca, Angelo Maria Minnella

**Affiliations:** 1Institute of Ophthalmology, Università Cattolica del Sacro Cuore, 00168 Rome, Italy; 2UOC di Oculistica, Fondazione Policlinico Universitario A. Gemelli-IRCCS, 00168 Rome, Italy; 3UOC di Neurologia, Fondazione Policlinico Universitario A. Gemelli IRCCS, 00168 Rome, Italy; 4Department of Ophthalmology, University of Turin, 10126 Turin, Italy; 5Department of Cardiovascular, Endocrine-Metabolic Diseases and Ageing, Istituto Superiore di Sanità, 00161 Rome, Italy; 6Institute of Neurology, Università Cattolica del Sacro Cuore, 00168 Rome, Italy; 7John Walton Muscular Dystrophy Research Centre, Newcastle Hospitals NHS Foundation Trusts, Newcastle University, Newcastle Upon Tyne NE1 3BZ, UK

**Keywords:** facioscapulohumeral muscular dystrophy (FSHD), retinal vasculopathy, optical coherence tomography-angiography (OCT-A), tortuosity index (TI), vessel density (VD), foveal avascular zone (FAZ), superficial capillary plexus (SCP), deep capillary plexus (DCP)

## Abstract

Facioscapulohumeral muscular dystrophy (FSHD) is a slowly progressive muscular dystrophy with a wide range of manifestations including retinal vasculopathy. This study aimed to analyse retinal vascular involvement in FSHD patients using fundus photographs and optical coherence tomography-angiography (OCT-A) scans, evaluated through artificial intelligence (AI). Thirty-three patients with a diagnosis of FSHD (mean age 50.4 ± 17.4 years) were retrospectively evaluated and neurological and ophthalmological data were collected. Increased tortuosity of the retinal arteries was qualitatively observed in 77% of the included eyes. The tortuosity index (TI), vessel density (VD), and foveal avascular zone (FAZ) area were calculated by processing OCT-A images through AI. The TI of the superficial capillary plexus (SCP) was increased (*p* < 0.001), while the TI of the deep capillary plexus (DCP) was decreased in FSHD patients in comparison to controls (*p* = 0.05). VD scores for both the SCP and the DCP results increased in FSHD patients (*p* = 0.0001 and *p* = 0.0004, respectively). With increasing age, VD and the total number of vascular branches showed a decrease (*p* = 0.008 and *p* < 0.001, respectively) in the SCP. A moderate correlation between VD and EcoRI fragment length was identified as well (r = 0.35, *p* = 0.048). For the DCP, a decreased FAZ area was found in FSHD patients in comparison to controls (t (53) = −6.89, *p* = 0.01). A better understanding of retinal vasculopathy through OCT-A can support some hypotheses on the disease pathogenesis and provide quantitative parameters potentially useful as disease biomarkers. In addition, our study validated the application of a complex toolchain of AI using both ImageJ and Matlab to OCT-A angiograms.

## 1. Introduction

Facioscapulohumeral muscular dystrophy (FSHD) is a slowly progressive muscular dystrophy with a distinctive pattern of skeletal muscle weakness and a wide range of disease severity [1]. Subjects show progressive loss of muscle mass and strength, as well as replacement with fat and connective tissue in selected muscle groups [2], often with an asynchronous and asymmetrical pattern [3,4].

As first revealed by Fitzsimons et al. in 1987 [5] and then confirmed by Padberg et al. in 1995 [6], a retinal vasculopathy is considered an established component of the FSHD phenotype [7]. Traditional ophthalmologic findings in FSHD include retinal vascular tortuosity and retinal vessel abnormalities on fluorescein angiography (FA) such as teleangectasia, microaneurysms, areas of capillary closure, and fluorescein leakage due to increased permeability. The leakage of plasma constituents can occasionally lead to exudative retinal detachment [5]. A secondary glaucoma due to neovascularization can develop [8]. However, retinal vascular changes in FSHD patients are often subclinical. Current guidelines advise referral to ophthalmological specialists for FSHD patients with visual complaints or with a severe form of the disease. However, data on the frequency of assessment and the techniques to be used for accurate ophthalmological monitoring in FSHD are lacking.

Minor retinal vascular alterations are undetectable with fundus examination; thus, fluorescein angiography (FA) is considered the gold standard for evaluating retinal vasculature [9]. However, FA is an invasive and time-consuming technique that allows the visualization of the superficial vascular plexus only [10], and dye leakage as well as haemorrhage or opacities can make the underlying retinal pathology undetectable. First adapted from optical coherence tomography (OCT), OCT-angiography (OCT-A) is a recently developed imaging technique that can non-invasively image all the layers of the retinal vasculature without dye injection by processing the motion of erythrocytes [11]. More specifically, OCT-A provides depth-resolved images of blood flow in the retina and choroid with a resolution level several times higher than that obtained with older forms of imaging [12], providing quantitative parameters such as the foveal avascular zone (FAZ) area and vessel density (VD). Despite these advantages, except for one study [13], updated information about ophthalmological findings in FSHD detected using OCT-A is missing.

In medicine and healthcare, artificial intelligence (AI) has been primarily applied to the field of medical image analysis, where it has shown robust diagnostic performance. Over the past few years, AI has similarly been applied to ocular imaging, mainly fundus photographs, OCT, and OCT-A. A better understanding of retinal vasculopathy through OCT-A and AI-based analysis of angiograms appears particularly promising since it can provide quantitative parameters potentially serving as disease biomarkers.

The aim of this study was to evaluate retinal vascular involvement in FSHD using colour fundus photography and swept-source OCT-A, analysed through artificial intelligence (AI).

## 2. Materials and Methods

This retrospective study was approved by the Ethics Committee/Institutional Review Board of the Catholic University. This research adhered to the tenets of the Declaration of Helsinki and informed consent was obtained from all patients after full and detailed explanation of the goals and procedures of the study. All the clinical and imaging data reported in this study were retrospectively re-evaluated. Recruitment was performed according to a collaboration protocol with the Department of Neurology of Università Cattolica del Sacro Cuore, Fondazione Policlinico Universitario Agostino Gemelli IRCCS.

### 2.1. Inclusion and Exclusion Criteria

Inclusion criteria were an established clinical diagnosis of FSHD type 1, confirmed by genetic testing (presence of a 4q35 BlnI resistant, p13-E11 EcoRI fragment whose length was <40 kb), and the possibility of obtaining good quality ocular imaging.

A pre-existing dataset of healthy controls was used for comparison.

### 2.2. Neurological Examination

Patients underwent a complete neurological examination inclusive of the clinical severity score (CSS) [14] to assess disease severity. The CSS is a 10-grade scale, which takes into account the extent of weakness in various muscular districts, considering the descending spread of symptoms from the face and shoulders to pelvic and leg muscles typical of FSHD. Higher scores were assigned to patients with involvement of pelvic and proximal lower limb muscles [14]. The CSS was then corrected for the patient’s age at examination (aCSS):Age-corrected CSS = ((CSS × 2)/age at examination) × 1000

Before dividing by the age at examination, the severity score is multiplied by two to generate whole numbers. Then, the outcome of this calculation is multiplied by 1000 to improve the interpretation of the results and visualization in graphs [15].

### 2.3. Ophthalmological Assessment

All patients underwent a full ophthalmologic evaluation including best corrected visual acuity (BCVA), anterior segment slit lamp biomicroscopy examination, tonometry, and fundus ophthalmoscopy after pupil dilation.

Colour fundus photography and 3 mm × 3 mm OCT-A scans (320 × 320 pixels, 24-bit RGB) of the superficial (layer 1) and deep capillary (layer 2) were acquired for each patient using a DRI Triton Swept-Source OCT device (Topcon, Tokyo, Japan). The level of segmentation for each capillary plexus was automatically provided by the instrument. To detect the superficial capillary plexus (SCP), the upper segmentation line was situated at 2.6 µm under the inner limiting membrane (ILM), whereas the lower segmentation line was located 15.6 µm under the junction between the inner plexiform layer (IPL) and inner nuclear layer (INL). To identify the deep capillary plexus (DCP), segmentation lines were placed 15.6 µm under the junction between the IPL and INL and 70.2 µm under the junction between the IPL and INL. In case of incorrect automatic segmentation, segmentation boundaries were manually adjusted.

Colour fundus photographs were qualitatively assessed by two independent graders, blinded for the patient characteristics, to score vessel tortuosity through a four-point grading scale (none, mild, moderate, or severe).

### 2.4. OCT-A Image Processing

Each enrolled subject had both eyes scanned. However, in order to provide statistical sample independence [16], data from only one eye, randomly selected for each subject, were included in the analysis.

OCT scans were preliminarily examined for the presence of artifacts and then processed to obtain quantitative parameters including vessel tortuosity, vessel density, FAZ area, and FAZ acircularity. Image processing was performed using a combination of Mathwork’s Matlab (MathWorks, Inc., Natick, MA, USA), Fiji [17], and other Fiji plugins as described in Figure 1. Before the image processing steps, all OCT scans were converted into grayscale. Matlab and ImageJ/Fiji integration was made possible using two other Fiji plugins [18,19]. The machine learning classification task was performed using Fiji’s Trainable Weka Segmentation plugin [20,21], a wrapper around a Java-based machine learning workbench called WEKA (Waikato environment for knowledge analysis), developed by the Machine Learning Group at the University of Waikato (Hamilton, New Zealand) [21]. We followed the procedure described by Goselink et al. [13] and Lee et al. [11]. Briefly, the training features selected were Gaussian blur, Sobel filter, Hessian, difference of Gaussian, and membrane projections (membrane thickness 1, membrane patch size 19, minimum sigma 1, maximum sigma 16). Training of the algorithm was performed on a randomly selected OCT-A image. Two distinct models were trained, one for the superficial retinal layer, and one for the deep retinal layer. Using the trained model(s), classification was performed for all the images in the dataset (patients and healthy subjects) using the FastRandomForest classifier, a multi-threaded version of random forest [22], initialized with 200 trees and 2 random features per node. The classifier’s output consists of a segmentation probability map highlighting the retinal structures detected as vessels. The probability map was converted into a binarized image in Matlab.

#### 2.4.1. Tortuosity Index (TI)

The binarized image was skeletonized (each white object in the binary image was converted to a single-pixel line) in Fiji, and the skeleton features (branch length, vertices positions, branch euclidean distance) calculated in Fiji were used to compute the tortuosity index, as defined in Lee et al. 2017 [11]. In detail, the actual length of each branch and the imaginary straight length between two branch nodes—points of connections—were marked and calculated. Then, vessel tortuosity was calculated as the sum of branch lengths divided by the sum of imaginary straight lines between branch nodes [11].
Vessel tortuosity = sum of actual branch lengths/sum of straight lengths between branch nodes.

#### 2.4.2. Vessel Density Score (VD Score)

From the binarized image, the VD score was calculated as a ratio of the number of pixels of the corresponding vascular tissue to the total number of pixels in the image, following the approach described in Minnella et al., 2019 [23].

#### 2.4.3. FAZ Area

FAZ area calculations were performed on the binarized image, with a morphological closing on the image itself using a single-disk structuring element of fixed size (20 pixels) [24]. Conversion from measurements expressed in pixels in metric lengths and areas was performed considering a pixel transverse size of 9.37 µm [25].

### 2.5. Statistical Analysis

All statistical calculations were performed using OriginLabs’ Origin Pro 2016. A *p*-value < 0.05 was considered statistically significant. Values were expressed as frequencies (%), mean ± standard deviation (SD), or median (interquartile range, IQR) as appropriate.

After checking for normality, a two-sample *t*-test was used to assess differences in patient and control measurements.

## 3. Results

### 3.1. Population

A total of 33 patients (15 males, 18 females, mean age 50.4 ± 17.4, ranging from 13 to 76 years) with a diagnosis of FSHD and 22 healthy subjects (8 males, 14 females, mean age 44.7 ± 11.3 years) were included in the analysis. A total of 66 eyes from the 33 patients were initially included in this study. All patients were clinically affected with a median of 3.5 points (range 1.5–5) on the 10-point CSS [14] and 148.1 points (±60.9, 46.2–333.3) on the aCSS [15]. The mean EcoRI fragment size was 22.3 kb (±6.1 SD) ranging from 10 to 35 kb. Clinical data are summarized in Table 1.

### 3.2. Ophthalmological Examination

The mean BCVA was 0.9 (decimal) ± 0.2 standard deviation (SD) and intraocular pressure (IOP) was within normal values in all the examined eyes.

#### 3.2.1. Colour Fundus Photography

Tortuosity of the retinal arteries was observed in 48 (71%) eyes: mild, moderate, and severe vascular tortuosity were found in 17, 25, and 6 eyes, respectively (Table 1, Figure 2)

#### 3.2.2. Optical Coherence Tomography Angiography

A random sampling was performed in order to select a single (left or right), random eye OCT-A from the 132 available patient scans.

For the deep layer, 33 patients (20 right eyes and 13 left eyes) and 22 healthy subjects (11 right eyes and 11 left eyes) were selected for the following analyses, while for the superficial layer, 33 patients (16 right eyes and 17 left eyes) and 22 healthy subjects (14 right eyes and 8 left eyes) were included for a total of 110 OCT-A scans.

### 3.3. Machine Learning Results

The machine learning method correctly identified the major vessels. A representative sample of the processing results for a patient and a control is shown in Figure 3.

#### 3.3.1. Tortuosity Index

The TI of the superficial layer (SCP) was increased in FSHD patients (mean 1.16 ± 0.01) in comparison to controls (mean 1.15 ± 0.01); (t (53) = 3.62, *p* < 0.001) (Figure 4A). The deep layer (DCP) showed a decrease (−0.07,) in the TI in FSHD (mean 1.17 ± 0.01) patients in comparison to controls (mean 1.24 ± 0.01) (t (53) = −23.5, *p* = 0.05) (Figure 4B).

However, although statistically significant, the interpretation of this last result should take into consideration that the reliability of vessel length calculations could not be visually assessed in the deep layer (Figure 5). No significant correlations were found between the TI and clinical parameters.

#### 3.3.2. Vessel Density Score

The VD score in the SCP (Figure 6A) was increased in FSHD patients (mean 38.03 ± 4.32) compared to normal (mean 25.40 ± 1.58) controls (t (53) = 13.10, *p* = 0.0001).

Similarly, the VD score in the DCP (Figure 6B) was increased in FSHD patients (mean 45.52 ± 2.37) compared to normal (mean 29.07 ± 1.88) controls (t (53) = 27.31, *p* = 0.0004). In addition, a significant correlation was found between age and vascular parameters. With increasing age, VD scores and the total number of vascular branches showed a decrease (r = −0.45, *p* = 0.008 and r = −0.51, *p* < 0.001, respectively) in the superficial layer, realistically for a progressive age-related vascular rarefaction. A moderate correlation between VD and EcoRI fragment length was identified as well (r = 0.35, *p* = 0.048).

#### 3.3.3. Foveal Avascular Zone

The FAZ was automatically delineated, and its area was calculated considering a pixel size of 9.375 µm. As an example, we show in Figure 7 some cases of FAZ calculations. For the SCP, no statistically significant differences were found between FSHD patients (mean 0.29 ± 0.12) and healthy controls in FAZ areas (mean 0.34 ± 0.13) (Figure 7A). FSHD patients showed a sex difference, with the FAZ area being larger in females than in males (0.33 mm^2^ vs. 0.22 mm^2^; t (31) = −3.2, *p* = 0.003), in SCP only. The FAZ area of the SCP showed a positive correlation with CSS (r = 0.55 *p* < 0.001). In the DCP, a decreased FAZ area (−0.39 mm^2^) was found in FSHD patients (mean 0.40 ± 0.16) in comparison to controls (mean 0.79 ± 0.26), t (53) = −6.89, *p* = 0.01) (Figure 7B). The results are summarized in Table 2.

## 4. Discussion

The present study analysed retinal vascular involvement in FSHD, using fundus photographs and swept-source OCTA, in order to refine the ophthalmological phenotype of FSHD subjects, both in qualitative and quantitative terms. In our study, fundus photographs showed a high prevalence of retinal arterial tortuosity (77%), confirming evidence in the literature [5,6,13]. However, these retinal vascular changes did not cause complaints or vision loss in any patient. The use of OCTA provided a more detailed insight into FSHD ophthalmological manifestations. The quantitative analysis of the TI, VD score, and FAZ area showed statistically significant differences between FSHD patients and controls. FSHD patients showed an increase in the TI of the SCP, a decrease in the TI in the DCP, an increase in the VD score of both SCP and DCP, and a decrease in the FAZ area in the DCP in comparison to controls. Interestingly, a gender difference was found in the FAZ area (SCP), with higher values in females. Currently, there is no consensus in the literature about the effects of gender on retinal vascularity. VD and perfusion density (PD) seem to be not affected by sex [26]. The sole parameter potentially affected by gender is the FAZ area, which is larger in females compared to males [27]. Our findings on the FAZ area in FSHD patients reflect those of the general population.

In the present study, we found that, with increasing age, VD and the total number of vascular branches showed a decrease in FSHD patients. Age can have an impact on vascular results, considering that elderly subjects usually present a vascular rarefaction of the capillary plexa. However, the FSHD population (tendentially older than controls) presented an increased VD both in the SCP and DCP.

Our data on the TI support what was found by Goselink et al. [13]: the absence of smooth muscle in the capillary vessel wall as opposed to retinal arterioles could provide a possible explanation for the TI increase in the superficial layers. Our results are instead novel regarding VD scores. A possible molecular link between FSHD pathophysiology and neoangiogenesis, plausibly responsible for the increase in VD, is provided by the upregulation of several genes involved in neovessel formation [28], and in particular of the Wnt/Frizzled signalling pathway in the skeletal muscle of patients with active disease [29]. Further studies will be needed to confirm the relevance of this or other molecular pathways to the reported retinal vascular proliferation [30].

The FAZ area decreasing in DCP was likely related to the VD increase. The FAZ area is inversely connected to the microcirculatory condition, as demonstrated by the FAZ area enlargement in diabetic retinopathy and retinal vascular occlusive diseases for the destruction of the vascular arcades [31]. In addition, our study is noteworthy for validating the application of a complex toolchain of AI using both ImageJ and Matlab to OCTA angiograms. This pipeline could be replicated and applied in other studies.

### Considerations of AI and Study Limitations

While the FAZ area and VD calculations have shown a very good tolerance to image quality and artefacts, TI figures should be interpreted with the greatest care. We strictly followed the approach by Goselink et al. [13] for the TI calculations, in order to be able to compare our results to previous works. However, we have already raised some concerns [32] about the reliability of this method on the SCP. Specifically, we noticed that TI calculation reliability may be affected by the quality of segmentation in a somehow unpredictable manner. Even more, regarding the DCP, we must highlight that the presence of a complex lattice of smaller capillaries turns into the detection of an extremely fragmented short vessel network, a situation where tortuosity computation by itself may be rendered meaningless. Moreover, in the skeletonization process, the lengths of the vessels are calculated. However, this process calculates the lengths of the “branches”, which are the lines between the dots along the vessels, and therefore does not calculate the total length of the vessels, as a human evaluator would do.

The relatively small sample size is also a partial limitation of this study. In addition, we included only FSHD type 1 patients, but exploring possible similarities or differences in ophthalmological characteristics of FSHD type 2 patients could be worthwhile.

Further studies would be needed to clarify the potential correlations between the quantitative OCTA parameters and the severity of the muscular disease, and to determine which patients may have or develop more severe disease [7]. Thus, OCTA could become an important tool for the routine ophthalmological evaluation of FSHD patients.

## 5. Conclusions

The importance of assessing ophthalmological abnormalities in FSHD is not restricted to the possibility of treatable vision loss. Looking at retinal vasculopathy as an integral part of FSHD opens new horizons, helping to understand the pathogenesis of the disease. OCTA is a non-invasive tool potentially useful to assess retinal vasculopathy and to provide promising disease biomarkers. The identification of new parameters potentially associated with prognosis appears pivotal in neuromuscular disorders such as FSHD where the extent and severity of involvement can vary enormously.

## Figures and Tables

**Figure 1 diagnostics-13-00982-f001:**
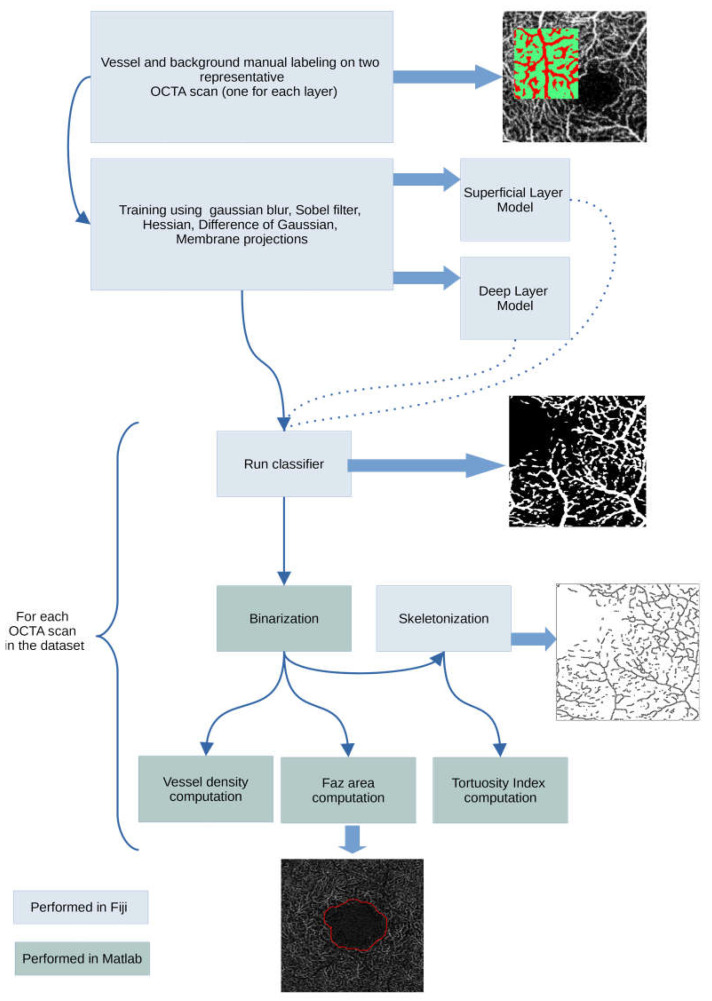
Image processing workflow.

**Figure 2 diagnostics-13-00982-f002:**
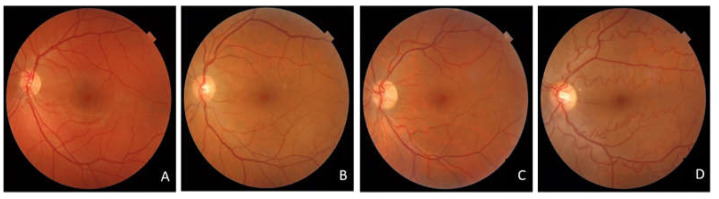
Colour fundus photography showing several grades of vascular tortuosity found in FSHD patients: (**A**) absent, (**B**) mild, (**C**) moderate, (**D**) severe vessel tortuosity.

**Figure 3 diagnostics-13-00982-f003:**
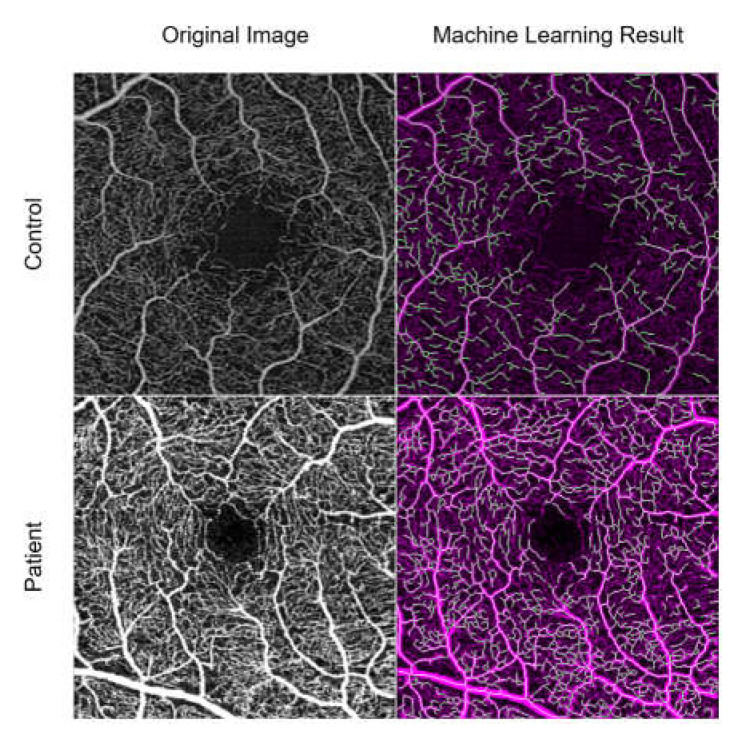
A representative sample of the machine learning processing for a control (**top**) and patient (**bottom**). On the left is the original OCT-A image, on the right is the binarization and skeletonization process.

**Figure 4 diagnostics-13-00982-f004:**
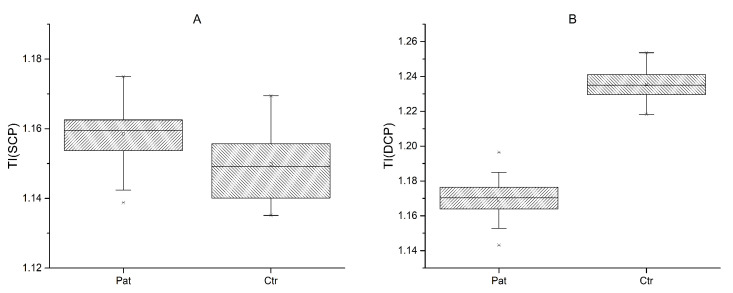
Tortuosity index (TI) in patients and normal controls for superficial (SCP) (**A**) and deep (DCP) (**B**) capillary plexus.

**Figure 5 diagnostics-13-00982-f005:**
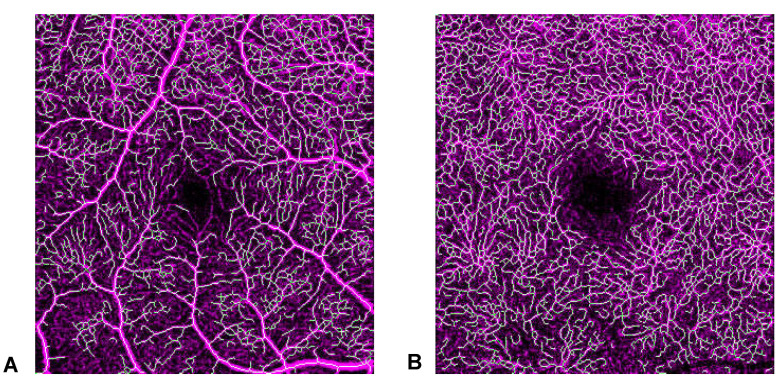
Skeletonization results from machine learning in superficial (SCP) (**A**) and deep (DCP) (**B**) capillary plexus. The reliability of the process can be visually assessed only for SCP.

**Figure 6 diagnostics-13-00982-f006:**
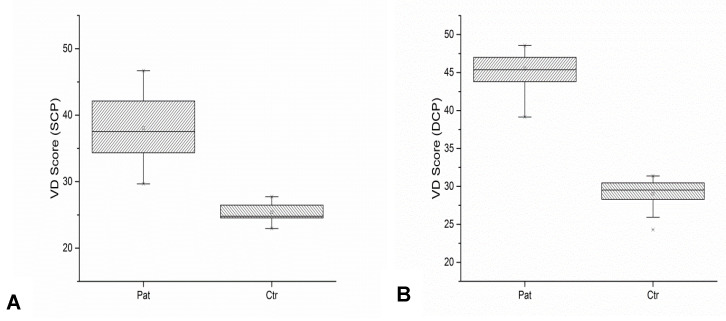
Vessel density (VD) scores in patients and normal controls for superficial (SCP) (**A**) and deep (DCP) (**B**) capillary plexus.

**Figure 7 diagnostics-13-00982-f007:**
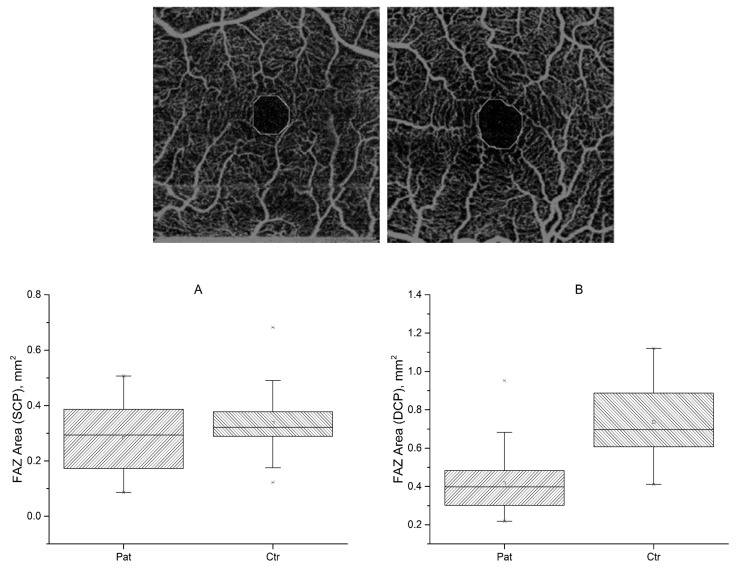
Top to bottom. Two examples of foveal avascular zone (FAZ) calculation. FAZ area for FSHD patients and normal controls, for superficial (**A**) (SCP) and deep (**B**) (DCP) plexuses.

**Table 1 diagnostics-13-00982-t001:** Demographic and clinical data: distribution of sex, EcoRI fragment length, age, clinical severity score (CSS), age-corrected CSS, and grade of retinal vascular tortuosity detected by colour fundus photography in FSHD patients.

Patients with FSHD (*N*)	33
Female (%), male (%)	54, 46
Age (years), mean (SD; range)	50.4 (17.4; 13–76)
EcoRI fragment (kB), mean (SD; range)	22.3 (6.1; 10–35)
CSS (0–10), median (IQR; range)	3.5 (1; 1.5–5)
aCSS, median (SD; range)	148.1 (60.9; 46.2–333.3)
Eyes with retinal arteries tortuosity (%)	71
Eyes with mild, moderate, severe tortuosity (%)	26, 38, 9

**Table 2 diagnostics-13-00982-t002:** Statistical analyses. * n.s.s.: not statistically significant.

	Tortuosity Index			Vessel Density Score			Faz Area		
	Controls	Patients		Controls	Patients		Controls	Patients	
	Mean ± SD		*p*	Mean ± SD		*p*	Mean ± SD		*p*
Superficial Capillary Plexus	1.15 ± 0.01	1.16 ± 0.01	<0.001	25.40 ± 1.58	38.03 ± 4.32	=0.0001	0.34 ± 0.13	0.29 ± 0.12	=0.123, n.s.s.*
Deep Capillary Plexus	1.24 ± 0.01	1.17 ± 0.01	=0.05	29.07 ± 1.88	45.52 ± 2.37	=0.0004	0.79 ± 0.26	0.40 ± 0.16	=0.01

## Data Availability

Data are available from the authors.

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
