# Peer review of "Artificial Intelligence for Evaluation of Retinal Vasculopathy in Facioscapulohumeral Dystrophy Using OCT Angiography: A Case Series"

_diagnostics, 2023, doi:10.3390/diagnostics13050982_

Round 1
Reviewer 1 Report
In this manuscript, the authors analyzed the retinal vascular involvement in patients with facioscapulohumeral muscular dystrophy (FSHD). The images were obtained by the color fundus photography and the OCT angiography. The artificial intelligence tool previously developed by other groups was used for the vascular binarization and skeletonization. Some parameters, such as the tortuosity index (TI), the vessel density (VD), and the foveal avascular zone (FAZ), were calculated and compared. The feasibility of the AI applications for retinal examination in FSHD was verified. The study is interesting. I have some suggestions that the authors may consider.
1. The framework of the AI model should be provided and described briefly.
2. What is the purpose of the fundus photography? Were the fundus images used for the FSHD diagnosis or the statistical analysis? Could either fundus photography or OCTA diagnose the FSHD? What are the features of FSHD in Figure 2? They should be described.
3. The detailed information of the patients in Table 1 is not necessary. The statistical information of the patients can be summarized in a table.
4. The AI tool was used for the OCTA segmentation. The segmentation accuracy should be evaluated.
5. There are many phases of “Error! Reference source not found” in the manuscript. They should be corrected. Please refer to Line 134, 205, etc.
Reviewer 2 Report
The study carried out, “Artificial Intelligence for evaluation of retinal vasculopathy in 2 Facioscapulohumeral Dystrophy using OCT angiography: a 3 case series” is interesting and necessary for the correct interpretation of the FSHD ophthalmological signs. The aims of the study are satisfied during the results, discussion and conclusion, ending that “The importance of assessing ophthalmological abnormalities in FSHD is not restricted to 361 the possibility of a treatable vision loss. Looking at retinal vasculopathy as an integral part 362 of FSHD opens up new horizons, helping to understand the pathogenesis of the disease. 363 OCTA is a non-invasive tool potentially useful to assess retinal vasculopathy and to pro-364 vide promising disease biomarkers. The identification of new parameters potentially as-365 sociated with prognosis appears pivotal in neuromuscular disorders such as FSHD where 366 extent and severity of involvement can vary enormously.” They said that more studies are needed, I agree but I consider that in those new future studies more participants, both male and female to equilibrate groups, are also needed to consolidate the results obtained in this study.
To improve the article, I propose the following details to the authors:
Abstract:
Line 19: change 33 to thirty-three.
Line 26 and 27: Put decimals with a point, not a comma.
The abstract is well written and summarizes what has been studied quite well. I suggest adding mean age of the participants.
I propose to add SCP and DCP as keywords.
Introduction:
Line 72: change “et. Al” by “et al.”
Comment something more about how artificial intelligence is being used in image analysis
Materials and methods:
Once Tortuosity Index (TI), Vessel Density 21 (VD) is defined, use it as an abbreviation throughout the text because it is used interchangeably in one way or another.
Line 134: reference error. Correct it, please.
I would recommend reinforcing how the images have been analyzed in the methods section, since it is a bit lacking.
Results:
There were differences in age and in number between males and females between groups, which could affect the results. What difficulties were found to expand the control sample and equalize the constant differences between age and the number of males and females?
Line 200 define standard deviation.
Line 205, 221, 229, 241: reference error.
Lines 228-232: change (Mean 228 1.16, SD 0.01) by (Mean 228 1.16 ± 0.01)
Figure 4: define SCP and DCP in the figure legend.
Figure 6: define VD, SCP and DCP in the figure legend.
Lines 249-258 and 273-280: change (Mean 25.40, SD 1.58) by (Mean 25.40 ± 1.58)
Line 270: use the abbreviation FAZ. “The FAZ was….”
Line 276: use DCP not deep plexus
Define FAZ in the legend of Figure 7.
Table 2: it is preferable to put the numerical value of p even if it is not <0.05
Discussion and conclusions:
During the discussion, the previously defined abbreviations SCP, DCP, VD, TI are not used either. In addition, capital letters are used interchangeably when naming these terms. This makes the discussion, although well written, lose formality. Please correct this throughout the article.
The text is justified in some paragraphs and not in others.
The discussion is consistent and compares the results obtained with other studies, but they do not talk aboutt the limitations that have occurred in the present study.
The conclusion is concise and related to the results obtained from this study.
Round 2
Reviewer 1 Report
All the points from my previous review have been addressed in satisfying accuracy. I believe that the manuscript is in good shape to be published in this journal.
Reviewer 2 Report
All revisions proposed to the authors have been satisfactorily added and corrected. I recommend accepting the article in this version.